# ArtHOI: Articulated Human-Object Interaction Synthesis via Dynamics Distillation

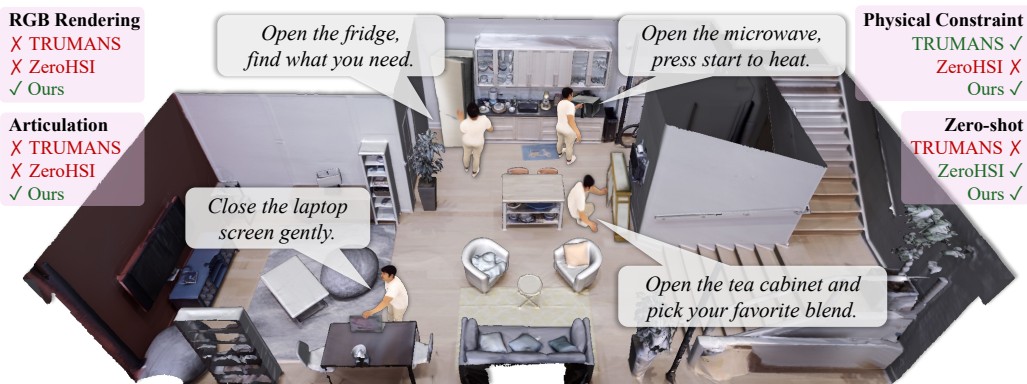

Figure 1: ArtHOI enables zero-shot synthesis of realistic articulated human-object interactions from text prompts. Unlike prior works (*e.g.*, TRUMANS, ZeroHSI), our method achieves all four capabilities simultaneously: RGB rendering, articulated object modeling, physical constraint modeling, and zero-shot generalization, notably without using 3D supervision.

## Abstract

Synthesizing realistic articulated human-object interactions is challenging, especially when explicit 3D/4D supervision is unavailable. Recent zero-shot methods distill dynamics priors from pretrained video diffusion models, but this setting inherently provides only monocular evidence. That makes articulated part motion highly ambiguous and tightly coupled with human actions, so prior work falls back to rigid-object assumptions and fails on everyday articulated scenes (*e.g.*, containing doors, fridges, cabinets). We introduce **ArtHOI**, the first zero-shot framework for synthesizing articulated human-object interactions via dynamics distillation from monocular video priors. We make two critical designs: **1)** *Flow-based part segmentation*: we use optical-flow cues to separate dynamic from static regions, because motion is the most reliable signal when multi-view information is absent. **2)** *Decoupled dynamics distillation*: joint optimization of human motion and object articulation is unstable under monocular ambiguity, so we first recover object articulation, then synthesize human motion conditioned on the reconstructed object states. ArtHOI distills dynamics from monocular 2D video priors without any 3D/4D ground truth. Across diverse scenes, ArtHOI yields physically plausible articulated interactions, improving contact quality and reducing penetration while enabling behaviors beyond rigid-only baselines. This extends zero-shot HOI synthesis from rigid manipulation to articulated dynamics. Code will be available.

## 1 Introduction

Synthesizing realistic human motions that interact with 3D environments is fundamental to computer graphics, VR/AR, embodied AI, and robotics applications (Li et al., 2024b; Kulkarni et al., 2024; Jiang et al., 2024a;b; Gao et al., 2020; Diller & Dai, 2024; Li et al., 2024c; Xu et al., 2023; Fan et al., 2025b). These interactions encompass both static scenarios (*e.g.*, sitting, lying) and dynamic

Table 1: Comparison of capabilities across different approaches. **HOI**: Human-object interaction; **RGB**: RGB rendering; **Art**: Articulated objects; **Phy**: Physical constraints; **ZS**: Zero-shot.

| Method | HOI | RGB | Art | Phy | ZS | Method | HOI | RGB | Art | Phy | ZS |
|---|---|---|---|---|---|---|---|---|---|---|---|
| CHOIS | ✓ | ✗ | ✗ | ✓ | ✗ | Nifty | ✓ | ✗ | ✗ | ✓ | ✗ |
| LINGO | ✓ | ✗ | ✗ | ✓ | ✗ | TRUMANS | ✓ | ✗ | ✗ | ✓ | ✗ |
| GenZi | ✓ | ✗ | ✗ | ✗ | ✓ | ZeroHSI | ✓ | ✗ | ✗ | ✗ | ✓ |
| InterDreamer | ✓ | ✗ | ✗ | ✗ | ✓ | Chao et al. | ✗ | ✗ | ✓ | ✓ | ✗ |
| Song et al. | ✗ | ✗ | ✓ | ✓ | ✗ | **Ours** | ✓ | ✓ | ✓ | ✓ | ✓ |

scenarios involving articulated objects (*e.g.*, opening doors, fridges, or cabinets). Such interactions are remarkably diverse and pervasive in daily life, involving numerous articulated objects and various interaction scenarios (Zhang & Lee, 2025; Zhang et al., 2025c). Despite significant advances in human or object motion synthesis (Zhu et al., 2023; Li et al., 2023; Lin et al., 2023; Petrovich et al., 2021; Xie et al., 2021; Xiao et al., 2025; Zhang et al., 2024; Fan et al., 2025a), synthesizing this broad spectrum of articulated human-object interactions remains a fundamental challenge due to the complex kinematic constraints and part-wise motion dependencies inherent in articulated objects.

This challenge stems from the prohibitive cost of collecting 3D/4D ground truth data, particularly for articulated human-object interactions. Prior work in human interaction synthesis primarily relies on datasets containing paired 3D scene and Mocap data (Jiang et al., 2024b; Hassan et al., 2021). While these methods can generate realistic motions for everyday activities such as navigation and sitting, they exhibit limited generalization across environments and interaction types due to the high cost and limited diversity of the Mocap collection. The acquisition of 3D/4D supervision for articulated interactions is particularly challenging, requiring precise tracking of both human motion and object part movements, specialized capture setups, and extensive manual annotation. Recent approaches have sought to circumvent this dependency by leveraging video generation models as motion priors (Li et al., 2024a; Xu et al., 2024; Li & Dai, 2024). For example, ZeroHSI (Li et al., 2024a) demonstrates that video diffusion models can be utilized to generate plausible 4D interactions through differentiable rendering, without requiring 3D/4D data. However, this zero-shot setting inherently operates on monocular evidence, which creates fundamental challenges for articulated object understanding.

However, these methods are inherently constrained to rigid object manipulation, treating dynamic objects as single entities undergoing global 6D transformations (Li et al., 2024b; Jiang et al., 2024a). They are incapable of modeling articulated objects where parts exhibit relative motion under kinematic constraints. This limitation stems from two fundamental challenges: 1) the difficulty of segmenting movable parts from monocular video; and 2) the tight coupling between human action and object articulation, which renders joint optimization unstable and prone to failure. As summarized in Table 1, no existing zero-shot method can generate interactions involving articulated object motion.

To address these limitations, we propose a novel framework that enables zero-shot synthesis of human interactions with articulated objects. Our key technique is to decouple the estimation of object articulation from human motion synthesis, avoiding the instability of joint optimization. Our approach introduces a structure-aware dynamics distillation pipeline that extracts articulated object dynamics from 2D video generation priors (KLING AI Team, 2024) without requiring 3D supervision. This eliminates the need for expensive 3D/4D data collection while leveraging the rich motion priors embedded in pretrained video diffusion models. Specifically, we first estimate part-wise articulated object motion using flow-based segmentation (Karaev et al., 2024; Teed & Deng, 2020) and articulation-aware regularization (Igarashi et al., 2005). Then, we refine human motion conditioned on the reconstructed articulated object states. This two-stage decoupled optimization strategy enables stable and physically plausible articulated interaction generation by separating object articulation from human motion synthesis. We demonstrate our approach on diverse scenes, generating plausible articulated human-object interactions that are beyond the reach of prior art.

Our main contributions can be summarized as follows: **First**, we present the first zero-shot framework for human interactions with articulated objects, extending video-prior-driven HOI beyond rigid manipulation. **Second**, we introduce a two-stage structure-aware dynamics distillation pipeline that recovers articulated object dynamics from 2D video priors without 3D supervision. **Third**, we demonstrate a decoupled optimization strategy that enables stable and physically plausible articulated interaction generation by separating object articulation from human motion synthesis.

## 2 RELATED WORK

**Human-Object Interaction Synthesis.** Synthesizing plausible human motions while manipulating objects has long been studied in computer animation, robotics, and embodied AI (Gao et al., 2020; Diller & Dai, 2024; Xu et al., 2023; Fan et al., 2025b). Traditional approaches rely on motion capture datasets paired with object trajectories (Kulkarni et al., 2024; Jiang et al., 2024b), enabling data-driven synthesis of interactions. Recent methods generate interactions from language prompts and sparse object waypoints, but require training on interaction-specific data (Li et al., 2024b; Jiang et al., 2024a; Zhang et al., 2025b) or assume known object kinematics (Huang et al., 2025; Li et al., 2025a; Jiang et al., 2023). These models exhibit limited generalization due to their dependence on curated motion sequences. In contrast, zero-shot methods circumvent this dependency by leveraging external priors. GenZi (Li & Dai, 2024) generates static human poses using 2D diffusion models, while ZeroHSI (Li et al., 2024a) synthesizes dynamic 3D human-object interactions from image-to-video models. However, existing zero-shot approaches assume only 6D rigid object manipulation.

**Articulated Object Modeling.** Modeling articulated objects requires understanding object kinematics. A significant line of work focuses on reconstructing articulated object structure and motion from visual inputs (Chao et al., 2025; Song et al., 2024; Zhai et al., 2025; Yao et al., 2025; Guo et al., 2025; Lin et al., 2025). While these methods can estimate part segmentation and trajectories, they often rely on category-level templates or known part hierarchies, limiting their applicability to novel objects. More recent approaches adopt unsupervised paradigms to discover articulated parts from motion cues alone (Deng et al., 2024; Peng et al., 2025; Liu et al., 2023; Xu, 2021; Goyal et al., 2025; Zhang et al., 2025d). However, these methods operate purely on object-centric motion and ignore the rich semantic and physical signals provided by human-object interaction.

**Video Distillation for 3D Reconstruction.** Recent zero-shot 3D methods leverage video diffusion models (VDMs) as powerful priors to generate 4D human-scene interactions without 3D supervision. Methods like Zero4D (Park et al., 2025) and Free4D (Liu et al., 2025a) show that a single input video can be extended into coherent 4D sequences by sampling from VDMs, while VideoScene (Wang et al., 2025) distills these outputs directly into 3D Gaussian representations in a single forward pass. Li et al. (Li et al., 2025b) specifically addresses articulated object kinematics by distilling motion patterns from video diffusion models, demonstrating the potential of VDMs for understanding articulated dynamics. Recent work has also explored diffusion-based generation of articulated objects (Zhang et al., 2025a; Kreber & Stueckler, 2025; Su et al., 2025; Gao et al., 2025). However, these methods model objects as monolithic entities with a single global transformation, failing to capture the part-wise articulation essential for human-object interaction.

## 3 METHODOLOGY

We address the problem of synthesizing realistic, articulated human-object interaction dynamics **without any 3D supervision**. Given a text prompt $\mathcal{T}$, our method outputs a temporally coherent 3D motion sequence involving a human (represented via SMPL-X (Pavlakos et al., 2019; Loper et al., 2015)) and an articulated object, both modeled using 3D Gaussians. Specifically, the human is parameterized by shape $\boldsymbol{\beta} \in \mathbb{R}^{10}$, pose $\boldsymbol{\psi}(t) \in \mathbb{R}^{J \times 3}$, and translation $\boldsymbol{\tau}(t) \in \mathbb{R}^{3}$, while the object parts are governed by $SE(3)$ transformations $\mathbf{T}^d(t)$. As illustrated in Fig. 2, we first generate a monocular video $\mathcal{V} = \{I(t)\}_{t=1}^{T}$ from $\mathcal{T}$ using video diffusion models. Then, we distill 3D dynamics from this 2D video through a **decoupled two-stage framework**: (1) In *Stage I* (Sec. 3.1), we identify articulated object parts using optical flow and SAM-guided segmentation, and recover their 3D articulation via differentiable rendering; (2) In *Stage II* (Sec. 3.2), we refine human motion conditioned on the reconstructed object dynamics, ensuring physically plausible contact and temporal coherence. This monocular-aware decoupling effectively resolves the ambiguity between human and object dynamics, enabling realistic interaction synthesis from monocular priors alone.

### 3.1 FLOW-BASED PART SEGMENTATION

We present a flow-based segmentation approach that leverages optical flow tracking and SAM-guided segmentation to identify articulated object components from the generated monocular video (Fig. 2 left). Given the monocular video sequence $\mathcal{V} = \{I(t)\}_{t=1}^{T}$ generated from a pre-trained video diffusion model based on the text prompt $\mathcal{T}$, we generate segmentation masks $M^h(t)$ for hu-

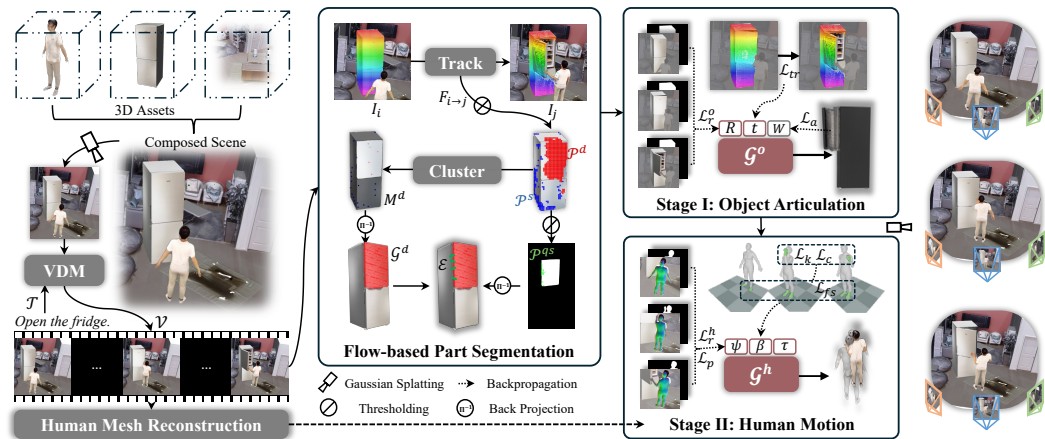

Figure 2: ArtHOI synthesizes articulated human-object interactions from monocular video priors. Given only synthesized 2D video $\mathcal{V}$ (no 3D supervision), we first recover object articulation using optical flow and differentiable rendering (Stage I), then synthesize human motion conditioned on the reconstructed object (Stage II). This monocular-aware decoupling resolves the ambiguity between human action and object dynamics, enabling realistic interactions with articulated objects.

man and $M^o(t)$ for object from each frame $I(t)$. To identify articulated parts, we leverage optical flow to provide crucial temporal motion cues that naturally distinguish between static and dynamic object components, as traditional segmentation methods struggle with the complex part-wise motion patterns of articulated objects under monocular observation. We use SAM-guided (Kirillov et al., 2023) segmentation with optical flow tracking (Karaev et al., 2024) and back projection (see Fig. 3 (a)) for robust articulated part identification.

**Optical Flow Tracking.** The segmentation process begins with optical flow tracking to identify dynamic and static regions (Fig. 2 middle top). For each frame pair $[I(t), I(t + 1)]$, we compute optical flow $F_{t \rightarrow t+1}$ using a pre-trained flow estimation network (Karaev et al., 2024). We recognize dynamic and static points by a threshold $\tau_f$:

$$\mathcal{P}^d = \{p \in M^o(t) \mid \|F_{t \rightarrow t+1}(p)\|_2 > \tau_f\}, \quad \mathcal{P}^s = \{p \in M^o(t) \mid \|F_{t \rightarrow t+1}(p)\|_2 \le \tau_f\}. \quad (1)$$

**SAM-guided Segmentation.** We then employ Segment Anything (SAM) (Kirillov et al., 2023; Ravi et al., 2024) to generate precise segmentation masks (Fig. 2 middle). Given the dynamic and static 2d point prompts, SAM produces a binary mask $M^d(t)$ that separates the articulated object parts:

$$M^d(t) = \text{SAM}(I(t), \mathcal{P}^d, \mathcal{P}^s) \quad (2)$$

**Back Projection.** To refine the segmentation and ensure 3D consistency, we project the SAM mask back to the 3D Gaussian space through back projection (Stearns et al., 2024) (see Fig. 3 (a)). This yields articulation weights $w_i^d$ and $w_i^s$ for the $i$-th Gaussian, indicating its dynamic and static contributions. We then define the dynamic and static Gaussian sets as $\mathcal{G}^d = \{\mathbf{g}_i \in \mathcal{G}^o \mid w_i^d > w_i^s\}$ and $\mathcal{G}^s = \{\mathbf{g}_i \in \mathcal{G}^o \mid w_i^s \ge w_i^d\}$.

To establish kinematic constraints between dynamic and static parts, we identify quasi-static pairs that represent potential articulation points (see Fig. 3 (b)). We identify quasi-static regions $\mathcal{P}^{qs} = \{p \in \mathcal{P}^d \mid \|F_{0 \rightarrow T}(p)\|_2 \le \tau_s \cdot \bar{F}_{0 \rightarrow T}\}$ from the dynamic regions where motion magnitude is relatively low, where $\bar{F}$ is the average motion magnitude across the entire sequence. We establish binding pairs $\mathcal{E}$ by connecting quasi-static Gaussians to their near static Gaussians within radius $r$:

$$\mathcal{E} = \{[\mathbf{g}^{qd}, \mathbf{g}^{qs}] \mid \mathbf{g}^{qs} \in \mathcal{G}^d, \Pi(\mathbf{g}^{qs}) \in \mathcal{P}^{qs}, \mathbf{g}^{qd} \in \mathcal{G}^s, \|\mathbf{g}^{qd} - \mathbf{g}^{qs}\|_2 \le r\}, \quad (3)$$

where $\mathbf{g}^{qs}/\mathbf{g}^{qd}$ represents the corresponding quasi-static/quasi-dynamic Gaussians, respectively.

### 3.2 DECOUPLED DYNAMICS DISTILLATION FRAMEWORK

Our framework employs a novel two-stage optimization strategy that decouples object articulation estimation from human motion synthesis to address the inherent ambiguity of monocular obser-

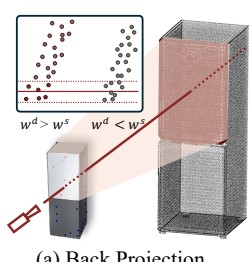 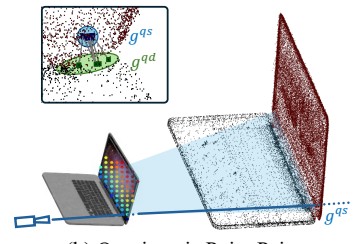 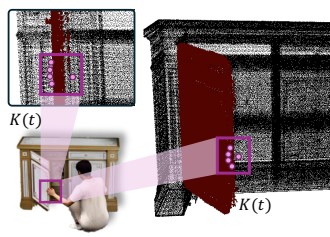

(a) Back Projection     (b) Quasi-static Point Pairs     (c) Human-object Contact Loss

Figure 3: Key components for articulated interaction under monocular supervision. (a) Back projection maps masks to 3D to identify moving parts. (b) Quasi-static point pairs link dynamic/static regions for kinematic stability. (c) Contact loss projects 2D keypoints into 3D using object depth, guiding human motion without multi-view cues. Ablations in Fig. 5 (middle: (b), right: (c)).

vations. This separation avoids the instability of joint optimization, while preserving the intricate coupling between human actions and object articulations, leading to more stable and accurate results.

**Stage I: Object Articulation.** We first focus on estimating the articulations of the segmented object parts, leveraging the flow-based segmentation to guide articulation estimation through optimization (Fig. 2 right top). We first focus on object articulation as it provides a more constrained optimization problem with clear kinematic constraints, making it easier to converge to physically plausible solutions and creating a stable reference frame for subsequent human motion synthesis.

We represent the articulated object motion using rotation matrix $\mathbf{R}^d(t) \in SO(3)$ and translation $\mathbf{t}^d(t) \in \mathbb{R}^3$, with transformation $\mathbf{T}^d(t) = \left[\mathbf{R}^d(t), \mathbf{t}^d(t)\right]$. We also introduce articulation weights $\mathbf{W}^o \in \mathbb{R}^{V \times J}$ for each part, determined from the flow-based segmentation results and fixed during optimization. The object Gaussians are driven by articulation parameters:

$$\boldsymbol{\mu}_i^o(t) = w_i^d \mathbf{T}^d(t) \boldsymbol{\mu}_i^o(0) + w_i^s \boldsymbol{\mu}_i^o(0) \tag{4}$$

where $w_i^d$ and $w_i^s$ are object articulation weights for the articulated part and static components, respectively. The optimization objective integrates multiple complementary constraints to ensure accurate and physically plausible articulation:

$$\min_{\{\mathbf{R}^d, \mathbf{t}^d\}} \mathcal{L}_r^o + \lambda_a \mathcal{L}_a + \lambda_s \mathcal{L}_s + \lambda_{tr} \mathcal{L}_{tr}. \tag{5}$$

The reconstruction loss $\mathcal{L}_r^o$ measures the alignment between the rendered object and the SAM masks:

$$\mathcal{L}_r^o = \|\mathcal{R}(\mathcal{G}^o(t)) - I(t)\|_2^2 + \beta^o \|\mathcal{S}(\mathcal{G}^o(t)) - M^o(t)\|_2^2. \tag{6}$$

The articulated loss $\mathcal{L}_a$ enforces kinematic constraints by maintaining the relative distances between quasi-static pairs, while the tracking loss $\mathcal{L}_{tr}$ aligns 3D Gaussian motion with 2D optical flow:

$$\mathcal{L}_a = \sum_{(\mathbf{g}^d, \mathbf{g}^s) \in \mathcal{E}} \|d(\mathbf{g}^d(t), \mathbf{g}^s(t)) - d(\mathbf{g}^d(0), \mathbf{g}^s(0))\|_2^2, \tag{7}$$

$$\mathcal{L}_{tr} = \sum_{\mathbf{g}^d \in \mathcal{G}^d} \|\Pi(\mathbf{g}^d(t)) - F_{t \to t+1}(\Pi(\mathbf{g}^d(t)))\|_2^2, \tag{8}$$

where $d(\cdot, \cdot)$ denotes the Euclidean distance, $\mathbf{g}^d(t)$ and $\mathbf{g}^s(t)$ are 3D positions of dynamic and static Gaussians at time $t$, and $\mathbf{g}^d(0)/\mathbf{g}^s(0)$ are their canonical positions. The tracking loss $\mathcal{L}_{tr}$ ensures that the 2D projection of 3D Gaussian points matches the corresponding 2D optical flow tracking, where $\Pi(\mathbf{g}^d(t))$ are 2D projections of 3D Gaussians and $F_{t \to t+1}(\Pi(\mathbf{g}^d(t)))$ are 2D tracked points from optical flow. The smoothness loss $\mathcal{L}_s$ regulates temporal consistency across the sequence.

**Stage II: Aligned Human Motion.** We then refine the human motion from the off-the-shelf human mesh reconstruction (HMR) (Shen et al., 2024) to align with the reconstructed object articulation from Stage I, leveraging the object articulation as constraints for human motion optimization (Fig. 2 right bottom). In this stage, we optimize the human motion parameters $\boldsymbol{\theta}(t)$ to drive the human

Gaussians $\mathcal{G}^h(t)$, while maintaining consistency with the object articulation estimated in Stage I. The optimization objective integrates multiple complementary constraints:

$$\min_{\boldsymbol{\theta}} \mathcal{L}_r^h + \lambda_p \mathcal{L}_p + \lambda_{fs} \mathcal{L}_{fs} + \lambda_s \mathcal{L}_s + \lambda_k \mathcal{L}_k. \tag{9}$$

The reconstruction loss $\mathcal{L}_r^h$ ensures visual alignment with the input video:

$$\mathcal{L}_r^h = \sum_{t=1}^{T} \|\mathcal{R}(\mathcal{G}^h(t)) - I(t)\|_2^2 + \beta^h \sum_{t=1}^{T} \|\mathcal{S}(\mathcal{G}^h(t)) - M^h(t)\|_2^2, \tag{10}$$

where $\mathcal{R}(\cdot)$ and $\mathcal{S}(\cdot)$ denote rendering and silhouette extraction respectively. The kinematic loss $\mathcal{L}_k$ leverages 3D contact keypoints derived from the object articulation (see Fig. 3 (c)):

$$\mathcal{L}_k = \sum_{t=1}^{T} \sum_{j \in \mathcal{K}_t} \|\mathbf{J}_j(\boldsymbol{\theta}(t)) - \mathbf{K}_j(t)\|_2^2, \tag{11}$$

where $\mathcal{K}_t$ is the set of confident contact keypoints at time $t$, $\mathbf{J}_j(\boldsymbol{\theta}(t))$ is the $j$-th joint position from SMPL-X, and $\mathbf{K}_j(t)$ is the corresponding 3D contact keypoint for the $j$-th joint.

Additionally, we incorporate the prior loss $\mathcal{L}_p$ to maintain natural human motion from the video diffusion model, and the foot sliding loss $\mathcal{L}_{fs}$ to prevent unrealistic foot movement during interactions:

$$\mathcal{L}_p = \sum_{t=1}^{T} \|\boldsymbol{\theta}(t) - \boldsymbol{\theta}_v(t)\|_2^2 + \eta \sum_{t=1}^{T} \|\boldsymbol{\psi} - \boldsymbol{\psi}_v(t)\|_2^2, \quad \mathcal{L}_{fs} = \sum_{t=1}^{T} \sum_{v \in \mathcal{V}_{foot}} \|\mathbf{v}(t) - \mathbf{v}(t-1)\|_2^2, \tag{12}$$

where $\boldsymbol{\theta}_v(t)$ and $\boldsymbol{\psi}_v(t)$ are the VDM-estimated parameters, and $\mathcal{V}_{foot}$ represents the foot vertices. The foot sliding loss leverages the foot contact estimation from GVHMR (Shen et al., 2024).

The final rendering combines all three Gaussian sets: $\mathcal{G}(t) = \mathcal{G}^h(t) \cup \mathcal{G}^o(t) \cup \mathcal{G}^s$. Detailed implementations are provided in the supplementary material (see Sec. A).

## 4 EXPERIMENTS

We conduct comprehensive experiments to evaluate our ArtHOI performance on zero-shot articulated human-object interaction synthesis. Our evaluation covers two main aspects: interaction quality and articulated object dynamics. We compare our approach against several state-of-the-art baselines and demonstrate significant improvements across multiple metrics.

**Baselines.** We compare our method against four representative approaches: TRUMANS (Jiang et al., 2024b), a mocap-based method requiring paired 3D scene and motion capture data; LINGO (Jiang et al., 2024a), a language-guided human motion synthesis approach; CHOIS (Li et al., 2024b), a contact-aware human-object interaction synthesis method; and ZeroHSI (Li et al., 2024a), a zero-shot method leveraging video diffusion models for rigid object interactions. Additionally, for articulated object dynamics, we compare against D3D-HOI (Xu et al., 2021) and 3DADN (Qian et al., 2022), which are designed explicitly for monocular articulated object estimation, providing a direct comparison of our monocular-aware approach.

**Datasets and Metrics. 1) For articulated object dynamics,** we use single-view videos rendered from scenes in the ArtGS dataset (Liu et al., 2025b) with ground truth annotations. **2) For human-object interaction**, we follow ZeroHSI (Li et al., 2024a) where each scene is annotated with natural language descriptions of human-scene interactions and corresponding initial positions. The scenes are from Replicate (Straub et al., 2019), with humans from XHumans (Shen et al., 2023) and objects generated by Trellis (Xiang et al., 2024). We employ two categories of metrics: interaction quality and articulated object dynamics. The interaction quality metrics include X-CLIP (Ni et al., 2022) for semantic alignment, Smoothness for motion temporal consistency, Foot Sliding for foot sliding detection, Contact% for contact percentage, and penetration errors (Penetration%) for physical plausibility. The articulated object dynamics metrics include rotation errors (Rot) for joint angle accuracy. Details of all evaluation metrics are provided in the supplementary material (see Sec. C).

**Implementation Details.** We implement our framework using PyTorch (Paszke et al., 2019) on NVIDIA A100 GPUs. The video diffusion model is based on KLing (KLING AI Team, 2024), and

Table 2: Comparison of interaction quality. Smoothness (↓) is best interpreted among zero-shot methods. Non-zero-shot high smoothness stems from minimal contact, not motion instability.

| Method | Zero-shot | X-CLIP ↑ | Smoothness ↓ | Foot Sliding ↓ | Contact% ↑ | Penetration% ↓ |
|---|---|---|---|---|---|---|
| TRUMANS | ✗ | 0.169 | 0.84 | 1.10 | 29.07 | 0.12 |
| LINGO | ✗ | 0.205 | **0.30** | 0.43 | 30.12 | 0.36 |
| CHOIS | ✗ | 0.111 | 0.64 | 1.17 | 39.72 | 0.09 |
| ZeroHSI | ✓ | 0.204 | 1.74 | 0.44 | 61.95 | 1.49 |
| Ours | ✓ | **0.244** | 0.87 | **0.31** | **75.64** | **0.08** |

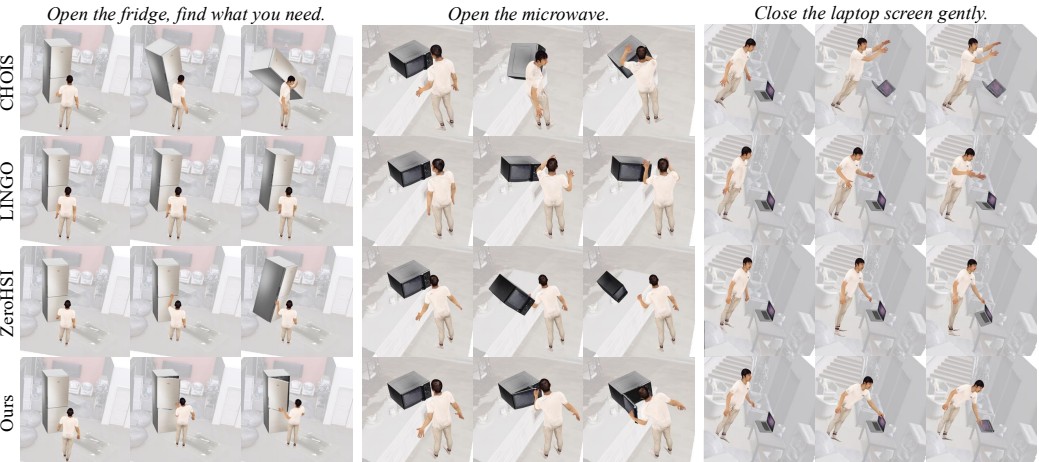

Figure 4: Qualitative comparison of our method with baselines. Our method synthesizes more realistic articulated human-object interactions with proper contact and natural motion coordination. Better inspected in our supplementary video.

we employ the Adam (Kingma & Ba, 2014) optimizer for training. We use 3D Gaussian splatting for differentiable rendering, enabling end-to-end optimization of both articulated object and human motions. Training typically takes approximately 30 minutes on a single NVIDIA A100 GPU.

## 4.1 INTERACTION QUALITY RESULTS

**Quantitative Comparison.** Table 2 presents the quantitative comparison of interaction quality metrics. Our method demonstrates superior performance across multiple key areas. We achieve the highest X-CLIP score (0.244), indicating superior semantic alignment between synthesized interactions and textual descriptions. In terms of foot sliding, our method achieves the lowest score (0.31), demonstrating more realistic foot contact during interactions. Most notably, we achieve the highest contact percentage (75.64%), showing that our method maintains more consistent human-object contact throughout the interaction sequence. While non-zero-shot methods (TRUMANS: 0.84, LINGO: 0.30, CHOIS: 0.64) achieve lower smoothness scores, these results should be interpreted with caution as *false positives*. The low smoothness scores of non-zero-shot methods stem from their minimal contact with articulated objects (Contact%: TRUMANS 29.07%, LINGO 30.12%, CHOIS 39.72%), rather than genuine motion stability. When human motion has limited interaction with objects, the smoothness metric can be artificially low due to the absence of complex contact dynamics that naturally introduce motion variations. In contrast, our method maintains competitive smoothness (0.87) while achieving significantly higher contact rates (75.64%), demonstrating our approach's ability to balance motion smoothness with realistic interaction complexity.

Our method achieves the lowest penetration errors (0.08), demonstrating superior physical plausibility compared to all baselines. This pattern correlates with the contact percentage results: our method achieves the highest contact percentage (75.64%), while baselines show lower contact rates (ZeroHSI: 61.95%, CHOIS: 39.72%). The superior penetration performance of our method demon-

strates the effectiveness of our flow-based segmentation and two-stage optimization in maintaining physically plausible interactions with articulated objects.

**Qualitative Comparisons.** Fig. 4 shows that our method generates realistic human interactions with diverse articulated objects. Data-driven baselines fail under complex prompts due to their lack of explicit articulated kinematics, while ZeroHSI treats objects as rigid and cannot model articulation. While ZeroHSI can generate human motion, it is fundamentally limited by its design and cannot achieve articulated object motion generation, treating all objects as rigid entities. In contrast, our method successfully handles complex articulated interactions by explicitly modeling object articulation through flow-based segmentation and two-stage optimization.

## 4.2 ARTICULATED OBJECT DYNAMICS RESULTS

Table 3 presents the comprehensive results for articulated object dynamics estimation. Our method demonstrates dramatically superior performance across all metrics compared to specialized methods. Our method achieves a mean rotation error of 6.71, representing a 73.3% reduction compared to D3D-HOI (25.13) and a 68.3% reduction compared to

Table 3: Articulated object dynamics metrics under **monocular** setting (without multi-view input).

| Method | Rot (mean) ↓ | Rot (max) ↓ | Rot (min) ↓ |
|---|---|---|---|
| D3D-HOI | 25.13 | 57.29 | 8.21 |
| 3DADN | 21.17 | 55.21 | 5.62 |
| Ours | **6.71** | **21.41** | **0.58** |

3DADN (21.17). Additionally, we achieve the lowest maximum rotation error (21.41 vs. 57.29 / 55.21) and minimum rotation error (0.58 vs. 8.21 / 5.62). These results validate our core contribution: the ability to accurately estimate articulated object dynamics from 2D video priors without requiring 3D supervision. The significant improvements in rotation estimation directly translate to more realistic and physically plausible articulated object motion during human interactions.

## 4.3 USER STUDY

To further validate the perceptual quality of our synthesized interactions, we conduct a comprehensive user study comparing our method with baseline approaches. We recruit 51 evaluators with diverse backgrounds to assess the quality of synthesized human-articulated object interactions. Participants rate results across four criteria: 1) **Realism** (naturalness and physical plausibility), 2) **Contact Quality** (accuracy of human-object contact), 3) **Motion Smoothness** (temporal coherence), and 4) **Overall Preference**. Full study details are in the supplementary material (Sec. B).

**Results.** Table 4 presents the comprehensive user study results across all evaluation dimensions. Our method demonstrates superior performance compared to all baseline approaches, with participants consistently preferring our synthesized interactions. Specifically, our method achieves the highest preference rates against TRUMANS (98.04% overall), CHOIS (95.28% overall), LINGO (91.51% overall), and ZeroHSI (89.42% overall). The results particularly highlight our method's strength in **Contact Quality** and **Motion Smoothness**, where we achieve 98.00% and 92.16% preference rates against TRUMANS, respectively. This validates that our flow-based segmentation and two-stage optimization effectively capture the complex dynamics of articulated human-object interactions, producing more realistic and temporally consistent results than existing approaches.

## 4.4 ABLATION STUDIES

We conduct extensive ablation studies to analyze the contribution of different components in our framework. Table 5 presents the quantitative results of removing individual components.

**Two-Stage Dynamics Distillation is crucial for stable performance.** The joint optimization shows intermediate performance compared to the full model, validating our decoupled strategy that separates object articulation estimation from human motion synthesis. As shown in Fig. 5 (a), the joint optimization approach (Joint Opt.) fails to learn reasonable interactions, as the coupled optimization of human motion and object articulation leads to unstable training dynamics and poor convergence.

**Articulation-aware regularization significantly improves articulated object dynamics.** Comparing rows 2 and 5, without articulation regularization ($\mathcal{L}_a$), the rotation errors increase dramatically (Rot (mean): $6.71 \rightarrow 15.67$), indicating that kinematic constraints are essential for maintaining

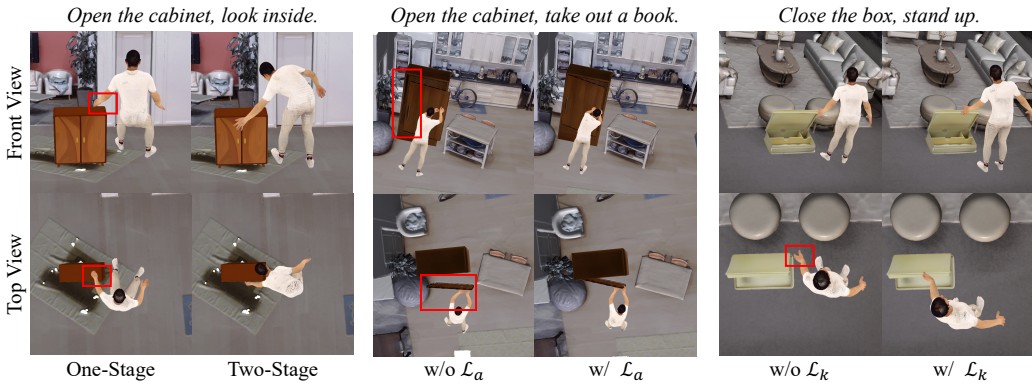

Figure 5: Comparing our full model with variants. Better inspected in our supplementary video.

Table 4: User study results showing the percentage of participants who preferred our method over each baseline across four evaluation dimensions.

| Method | Realism ↑ | Contact Quality ↑ | Motion Smoothness ↑ | Overall Preference ↑ |
|---|---|---|---|---|
| Ours vs. TRUMANS | 96.08% | 98.00% | 92.16% | 98.04% |
| Ours vs. CHOIS | 95.20% | 89.08% | 94.83% | 95.28% |
| Ours vs. LINGO | 90.20% | 87.13% | 92.00% | 91.51% |
| Ours vs. ZeroHSI | 91.18% | 85.41% | 84.21% | 89.42% |

Table 5: Ablation study results. We systematically remove individual components and evaluate their impact on both interaction quality and articulation accuracy.

| Method | Interaction | | | Articulation | | |
|---|---|---|---|---|---|---|
| | X-CLIP ↑ | Foot Sliding ↓ | Contact% ↑ | Rot (mean) ↓ | Rot (max) ↓ | Rot (min) ↓ |
| Joint Opt. | 0.187 | 0.67 | 61.45 | 12.34 | 35.89 | 2.01 |
| w/o $\mathcal{L}_a$ | 0.223 | 0.42 | 68.75 | 15.67 | 42.18 | 4.56 |
| w/o $\mathcal{L}_k$ | 0.201 | 0.58 | 59.82 | **6.71** | **21.41** | **0.58** |
| w/o $\mathcal{L}_s$ | 0.218 | 0.49 | 65.43 | 8.23 | 25.45 | 0.79 |
| Full Model | **0.244** | **0.31** | **75.64** | **6.71** | **21.41** | **0.58** |

physically plausible articulated object motion. The kinematic loss ($\mathcal{L}_k$) in row 3 and smoothness loss ($\mathcal{L}_s$) in row 4 further contribute to stable and coherent articulated motion generation. Fig. 5 (b) and (c) provide visual evidence: without articulation regularization ($\mathcal{L}_a$), the articulated parts of objects tend to separate from the main body, violating physical constraints and resulting in unrealistic object configurations. Most critically, removing kinematic loss ($\mathcal{L}_k$) severely degrades the quality of 3D hand-object interactions, as the model cannot maintain proper spatial relationships between human hands and articulated object parts during complex manipulation tasks.

## 5 CONCLUSION

This work introduces the first zero-shot framework for articulated human-object interaction synthesis via dynamics distillation from monocular video priors, addressing the limitation of existing methods that are inherently constrained to rigid object manipulation. Our key insight is to decouple object articulation estimation from human motion synthesis through flow-based segmentation and two-stage optimization, enabling stable reconstruction from monocular 2D video priors without requiring 3D supervision. Through comprehensive evaluation on diverse interaction scenarios, our approach achieves significant improvements over existing methods, generating realistic interactions with articulated objects that were previously out of reach. Our work extends the scope of zero-shot interaction synthesis beyond rigid manipulation to kinematically constrained environments, opening new possibilities for more realistic and scalable interaction synthesis in virtual reality and robotics.

ETHICS STATEMENT

This work does not directly involve human subjects. However, some publicly available datasets used in this study (*e.g.*, XHumans) may contain motion data derived initially from real humans, though fully anonymized and widely adopted in the research community. No personally identifiable information is used. All experiments were conducted using standard computational resources without environmental or societal harm. The methodology does not introduce discriminatory biases, and the model's potential applications are aligned with responsible AI principles. The authors have reviewed the ICLR Code of Ethics and confirm that this submission adheres to its guidelines.

REPRODUCIBILITY STATEMENT

To support reproducibility, we provide a complete description of our model architecture, training procedures, hyperparameters, and evaluation protocols in the main paper. Additional implementation details, including data preprocessing steps, optimization settings, and environment specifications, are included in the Appendix. We have strived to document all necessary components with sufficient clarity and precision to enable independent replication of our results.

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

## LLMs Usage

We acknowledge large language models (LLMs) in the preparation of this manuscript. Specifically, we utilized LLMs for text polishing, grammar correction, and improving the clarity. The core experimental results and scientific contributions remain entirely our own work.

## A   More Implementation Details

**Flow-based Segmentation.** Our flow-based segmentation approach leverages optical flow tracking and SAM-guided segmentation to identify articulated object parts from monocular video sequences. We use CoTracker (Karaev et al., 2024) for robust optical flow estimation, which provides temporal consistency across frames. The flow magnitude thresholds are set to 5 pixels for dynamic regions and 2 pixels for static areas to distinguish between moving and stationary parts. For SAM segmentation, we utilize the ViT-H model (Kirillov et al., 2023; Ravi et al., 2024) with default parameters, which provides precise boundary detection for articulated parts.

The quasi-static point identification uses a dynamic threshold based on the 10th percentile of motion magnitudes, with a minimum threshold of $1.0$ pixels to identify regions with minimal motion within dynamic areas. The binding radius $r$ is set to $0.05$ meters in 3D space, ensuring the proper kinematic constraint between connected parts.

**Two-Stage Optimization.** Our two-stage optimization framework employs different learning rates and optimization strategies for each stage. In Stage I (Object Articulation), we use Adam optimizer (Kingma & Ba, 2014) with learning rate $1.0 \times 10^{-4}$ for transformation parameters $\mathbf{T}^d(t)$. The loss weights are set as: $\lambda_r = 1.0$, $\lambda_{tr} = 2.0$, $\lambda_a = 0.05$, and $\lambda_s = 1.0$. The optimization runs for 200 iterations per frame with early stopping based on reconstruction loss convergence.

In Stage II (Human Motion), we use Adam optimizer (Kingma & Ba, 2014) with learning rate $1.0 \times 10^{-3}$ for body pose parameters and $1.0 \times 10^{-4}$ for camera parameters. The loss weights are: $\lambda_s = 1.0 \times 10^4$, $\lambda_k = 1.0 \times 10^4$ (kinematic loss), $\lambda_p = 1.0$, $\lambda_{fs} = 10$, and $\lambda_c = 1.0 \times 10^5$. The optimization runs for 1000 iterations, with contact loss being the primary driving force for realistic human-object interactions.

**3D Gaussian Splatting Integration.** We employ 3D Gaussian splatting (Kerbl et al., 2023) for differentiable rendering, enabling end-to-end optimization. The canonical Gaussians are initialized using the first frame of the input video sequence. For human representation, we use canonical 3D Gaussians $\mathcal{G}^h(0)$ distributed across the SMPL-X (Pavlakos et al., 2019; Loper et al., 2015) mesh surface. For articulated objects, we use canonical Gaussians $\mathcal{G}^o(0)$ per part, with articulation weights computed from the flow-based segmentation results.

**Training Setup.** All experiments are conducted on NVIDIA A100 (40GB) GPUs. The video diffusion model is based on KLing (KLING AI Team, 2024) with default parameters. The total training time is approximately 30 minutes per scene, including both stages of optimization. The batch size is set to 1 for memory efficiency, and we use gradient clipping with a maximum norm of $1.0$ to ensure training stability. The default number of iterations is set to 200 per frame for Stage I, with Stage II running for 1000 iterations total.

## B   Details of User Study

We conducted a comprehensive user study to evaluate the perceptual quality of our synthesized articulated human-object interactions. The study involved 51 participants with diverse backgrounds in computer graphics, robotics, and general technology. Each participant evaluated 20 interaction sequences across different types of articulated objects (doors, cabinets, and fridges).

**Evaluation Protocol.** Participants were presented with side-by-side comparisons of our method against baseline approaches (TRUMANS, CHOIS, LINGO, and ZeroHSI). For each comparison, they were asked to evaluate four criteria: **1) Realism.** How natural and physically plausible the human-object interactions appear, considering both human motion and object articulation. **2) Contact Quality.** The accuracy and consistency of contact between human body parts and articulated objects, including proper hand-object grasping and body-object support. **3) Motion Smoothness.**

The temporal consistency and fluidity of both human and object motion, without abrupt changes or unrealistic movements. **4) Overall Preference.** General preference ranking among different methods, considering all aspects of the interaction quality.

## C    DETAILS OF EVALUATION METRICS

**X-CLIP Score** measures semantic alignment between synthesized interactions and textual descriptions using cross-modal similarity (Ni et al., 2022). We use the X-CLIP to compute similarity scores between video frames and text prompts. The model processes $8$ sampled frames per video sequence with a frame sample rate of $1.0$, and we report the softmax probability corresponding to the correct scene description. Higher scores indicate improved text-to-motion correspondence.

**Motion Smoothness** evaluates temporal consistency by computing velocity stability and acceleration magnitude across human joint trajectories. We calculate joint velocities (first-order derivatives) and accelerations (second-order derivatives) for all SMPL-X (Pavlakos et al., 2019; Loper et al., 2015) joints at 30 FPS. The smoothness score is computed as the standard deviation of joint speeds across all frames, with lower values indicating smoother motion. We also report velocity stability and acceleration magnitude for comprehensive motion analysis.

**Foot Sliding** detects unrealistic foot movement during interactions using an advanced mesh-based algorithm. We analyze four foot joints (left/right ankles and toes) from SMPL-X (Pavlakos et al., 2019; Loper et al., 2015), computing their distances to the ground mesh and projecting displacements onto horizontal planes perpendicular to ground normals. The sliding threshold is set to $0.001$ m/frame, and we report the sliding score as the ratio of sliding frames to contact frames multiplied by average sliding distance. Lower values indicate more realistic foot contact.

**Contact Percentage** measures the percentage of frames where human body parts maintain proper contact with articulated objects. We compute distances between hand joints (left/right wrists) and object vertices. The metric reports both contact percentage and average contact distance, with higher contact percentages indicating more consistent interaction.

**Penetration Errors** quantifies physical plausibility using a mesh-based penetration detection algorithm. We compute distances between human vertices and scene/object meshes, using vertex normals to determine penetration direction. The penetration threshold is set to $0.3$ m, and we report penetration percentage (Penetration%).

**Rotation Errors** measures the angular difference between estimated and ground truth joint rotations using two evaluation protocols: (1) ART3D evaluation for 3D articulated object tracking, and (2) RT evaluation for rotation-translation estimation. We report mean, standard deviation, maximum, minimum, and median rotation errors across all joints and frames in degrees.

## D    DISCUSSION

**Runtime Analysis.** Our total runtime is composed of four main steps: video generation with KLing (5 min), flow-based segmentation (2 min), Stage I object articulation optimization (15 min), and Stage II human motion synthesis (8 min). The total runtime of 30 minutes on a single NVIDIA A100 GPU. This efficiency is achieved through our two-stage optimization strategy, where Stage I focuses on object articulation while Stage II synthesizes human motion, allowing for parallel processing opportunities in future implementations.

**Clarification on Zero-shot.** Our "zero-shot" approach means no fine-tuning is applied to foundation models (e.g., KLing video diffusion model) during interaction synthesis. This efficient design avoids the expensive cost of model adaptation while maintaining high-quality results. The optimization is scene-specific and runs on a single GPU within 30 minutes, making it practical for real-world applications.

**Practical Applications.** The practical implications of our work extend to multiple domains. In robotics, our method can generate training data for manipulation tasks involving articulated objects. In virtual reality and gaming, it enables the creation of realistic human-object interactions without extensive motion capture. The efficiency of our approach makes it suitable for real-time applications and rapid prototyping of interaction scenarios.

**Failure Cases.** We identify several failure cases in our current approach:

**1) Optical Flow Tracking Failures:** Co-tracker struggles with low-texture or reflective regions, leading to distortions that propagate into our articulation prediction results. When the articulated object surfaces lack sufficient visual features or contain specular reflections, the optical flow estimation becomes unreliable, resulting in incorrect motion tracking and subsequent failures in flow-based segmentation and Stage I (see Fig. A). **2) Complex Articulated Structures:** Our method struggles with objects having multiple degrees of freedom or non-rigid articulations (*e.g.*, soft-body joints, elastic

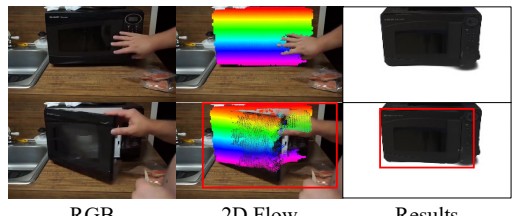

RGB       2D Flow       Results

Figure A: Co-tracker struggles with low-texture or reflective regions, leading to distortions that propagate into our articulation prediction results.

connections). The current kinematic constraints are designed for simple rotational and translational motions. **3) Long-term Temporal Consistency:** As video sequences become longer, cumulative errors in articulated object motion extraction can lead to gradual deviation from physical plausibility. The contact states between humans and objects may become unstable over extended interactions.

## E LIMITATIONS AND FUTURE WORK

**Limitations.** Despite significant improvements demonstrated in our experiments, our method has several limitations that warrant discussion: **1) Optical Flow Tracking Failures:** Co-tracker struggles with low-texture or reflective regions, leading to distortions that propagate into our articulation prediction results. When the articulated object surfaces lack sufficient visual features or contain specular reflections, the optical flow estimation becomes unreliable, resulting in incorrect motion tracking and subsequent failures in flow-based segmentation and Stage I (see Fig. A). This issue is particularly pronounced with metallic surfaces, glass objects, or uniform-colored surfaces where optical flow cannot establish reliable correspondences. **2) Complex Articulated Structures:** Our method struggles with objects having multiple degrees of freedom or non-rigid articulations (*e.g.*, soft-body joints, elastic connections). The current kinematic constraints are designed for simple rotational and translational motions, limiting our ability to handle sophisticated mechanical systems. Objects with multiple interconnected joints (such as robotic arms with 6+ DOF) or flexible components (like cables, hoses, or deformable materials) cannot be accurately modeled with our current rigid-body assumptions. **3) Long-term Temporal Consistency:** As video sequences become longer, cumulative errors in articulated object motion extraction can lead to gradual deviation from physical plausibility. The contact states between humans and objects may become unstable over extended interactions. Minor errors in early frames compound over time, causing objects to appear to "float" or penetrate through surfaces in sequences longer than $10 - 15$ seconds.

**Future Work.** We identify several promising directions for future research:

**1) Multi-DOF Articulation Modeling.** Extending our framework to handle complex articulated structures with multiple degrees of freedom, including soft-body dynamics, elastic connections, and non-rigid articulations. This would involve developing more sophisticated kinematic constraint models and optimization strategies that can handle objects with 6+ DOF, such as robotic arms and complex mechanical systems.

**2) Multi-object Interactions.** Extending the framework to handle complex scenarios involving multiple articulated objects and human interactions, including developing multi-object optimization strategies and interaction modeling techniques. This includes scenarios where humans interact with multiple objects simultaneously (e.g., opening a cabinet while holding a tool) or where objects interact with each other through human manipulation.

**3) Dataset and Benchmark Development.** Creating comprehensive datasets and standardized benchmarks for articulated human-object interaction synthesis to facilitate reproducible research and systematic method comparison. This would encompass diverse interaction scenarios across a wide range of articulated object categories (*e.g.*, doors, drawers, and tools), varied human actions (*e.g.*, opening, closing, pulling, and fine manipulation), and multiple environmental conditions.

