# OpenReview forum: "ArtHOI: Articulated Human-Object Interaction Synthesis via Dynamics Distillation"
_ICLR.cc/2026/Conference — ICLR 2026 Conference Withdrawn Submission_

### Official Review · Reviewer_hKKQ · 2025-10-28

**Soundness:** 2
**Presentation:** 3
**Contribution:** 2
**Rating:** 4
**Confidence:** 4

**Summary:**

In this paper, the authors introduce a zero-shot framework for synthesizing articulated human-object interactions using monocular video priors, called ArtHOI. It decouples object articulation from human motion synthesis to address instability in joint optimization. The method uses optical flow and SAM-guided segmentation to identify dynamic object parts, reconstructs their motion via differentiable rendering, and refines human motion based on contact-aware constraints. Authors claim that ArtHOI method achieves realistic interaction quality without 3D/4D supervision, outperforming prior rigid-only baselines in semantic alignment, contact accuracy, and physical plausibility.

**Strengths:**

1. The proposed framework has the capability to achieve zero-shot synthesis without relying on 3D/4D supervision, making it scalable and data-efficient.
2. The use of optical flow combined with SAM-guided segmentation enables identification of articulated object parts. This hybrid approach is not novel but it leverages motion cues and semantic priors to overcome limitations of traditional segmentation methods.
3. Decoupled dynamics distillation separating object articulation from human motion for stable optimization.

**Weaknesses:**

1. The segmentation pipeline in this paper heavily relies on SAM and optical flow, which may fail under occlusion or fast motion. The authors should justify the robustness of their approach across diverse scenes and consider evaluating alternative segmentation methods. Ablation studies are also missing and should be included.
2. All evaluations are conducted on synthetic data, with no real-world human-object interactions benchmarks results. The authors are encouraged to evaluate on BEHAVE’s [1] real RGB-D sequences using monocular input and report corresponding performance to demonstrate real-world generalization.
3. The method is tested on a narrow set of articulated objects. The authors should expand the evaluation to include more diverse object categories and interaction types. The OMOMO [2] and BEHAVE [1] datasets are particularly well-suited for this purpose, given their object-driven motion and inherent monocular ambiguity.
4. Gaussian splatting may oversimplify fine-grained geometry, especially for complex object surfaces. A comparison with mesh-based or hybrid representations would help clarify trade-offs between geometric fidelity, performance, and efficiency.
5. Contact keypoints are inferred heuristically without learning-based validation or annotation. The authors should justify the reliability of these estimates and consider integrating learned contact prediction. Comparing heuristic keypoints to BEHAVE’s [1] annotated contacts and measuring precision, recall, and downstream pose quality, would strengthen this analysis.
6. The paper uses smoothness as a stability metric; however, smoothness alone can be misleading, especially across varying interaction complexities. The authors should normalize smoothness by contact count or the number of moving parts, and propose additional metrics such as contact jitter or interpenetration volume. OMOMO [2] and BEHAVE [1] contain sequences with diverse contact patterns, enabling a more nuanced analysis of stability.
7. The segmentation pipeline’s novelty remains unclear, as it reuses SAM and optical flow without clearly articulating what is new. The integration appears incremental, and the paper lacks ablations to justify its design choices. The authors should describe how flow-guided SAM prompts are constructed and compare naive stacking with their integrated pipeline. Evaluating downstream performance on [1-2] would demonstrate whether the proposed ArtHOI provides real-world improvements.

[1] Behave: Dataset and method for tracking human object interactions, CVPR 2022

[2] Object Motion Guided Human Motion Synthesis, SIGGRAPH 2023

**Questions:**

1. The novelty in the segmentation pipeline beyond combining SAM and optical flow is limited? How does the proposed integration improve over naive stacking, and what ablation results support this claim on benchmarks like OMOMO and BEHAVE?
2. Does the proposed method generalize beyond synthetic data? How does it perform on real monocular benchmarks such as BEHAVE?
3. Can the method handle a wider range of articulated objects and interaction types? How does it perform on more diverse datasets like OMOMO and BEHAVE that include varied object-driven motions?

---

### Official Review · Reviewer_YiKG · 2025-10-29

**Soundness:** 3
**Presentation:** 3
**Contribution:** 2
**Rating:** 4
**Confidence:** 2

**Summary:**

In this paper, it proposed a HOI synthesis method, termed as ArtHOI. It utilize a video diffusion model to generate a HOI video. Then, it utilize part segmentation for the object. Finally, the articulated object and human motion is reconstructed from the HOI video. The proposed method has been validated in different scenes and shows better performance than some previous methods.

**Strengths:**

1. This paper utilize video diffusion motion for human-object interaction synthesis, making the synthesized HOI motion corresponding to video prior.
2. The articulated object can be modeled according to the video and text description.
3. The HOI synthesis is accompanied by human 3DGS modeling.

**Weaknesses:**

1. Articulated object modeling and motion capture has been widely researched. In this paper, it is utilized for HOI synthesis through a video diffusion model. This may diminish the significance of your contribution.
2. Rather than HOI synthesis, it is more like 4D reconstruction after a video diffusion model.
3. According to the video, the refrigerator is completely suspended in mid-air.
4. There no detailed description of the datasets for evaluation.

**Questions:**

1. The quantitative analysis is insufficient. Whether it is possible to compare the proposed method with CHOIS according to the experimental settings of CHOIS.

---

### Official Review · Reviewer_KEGa · 2025-10-29

**Soundness:** 2
**Presentation:** 3
**Contribution:** 3
**Rating:** 4
**Confidence:** 3

**Summary:**

This paper proposes ArtHOI, a zero-shot method for 3D human–articulated object interaction without 3D supervision, using optical flow + SAM part segmentation and a two-stage optimization to improve motion realism and physical plausibility over baselines.

**Strengths:**

Tackles an underexplored but important problem: zero-shot articulated human–object interaction generation in monocular settings.

Methodologically sound design: decoupling object and human motion yields improved stability and plausibility.

Comprehensive evaluation: quantitative benchmarks, ablations, and a user study.

**Weaknesses:**

1. Optical flow dependence: segmentation accuracy and articulation estimation degrade on low-texture or reflective surfaces, common in real-world scenarios, yet robustness was not quantitatively evaluated.

2. Restricted articulation types: currently supports only simple rotational/translational joints, excluding multi-DOF mechanisms and non-rigid structures, which limits applicability in advanced interaction contexts.

3. Long-horizon instability: sequences exceeding ~10–15 s exhibit noticeable drift and unstable contact; no remedies or systematic measurements are provided.

4. Efficiency constraints: approximately 30 min optimization per scene may be prohibitive for large-scale or near-real-time deployment.

**Questions:**

1. Could multi-modal cues (e.g. monocular depth estimation, self-supervised keypoint tracking) be integrated to improve segmentation robustness under low-texture/high-reflection conditions?

2. For multi-DOF or deformable objects, have the authors considered extending to multi-stage optimization or learning an articulation graph to capture complex kinematics?

3. Is long-sequence drift primarily due to object pose accumulation errors or misalignment between human motion and object state?

4. Different causes may require different mitigation strategies.
Can efficiency be improved via keyframe selection and interpolation, reducing optimization load without sacrificing temporal coherence?

---

### Official Review · Reviewer_xemN · 2025-10-31

**Soundness:** 3
**Presentation:** 3
**Contribution:** 3
**Rating:** 6
**Confidence:** 3

**Summary:**

This paper proposes ArtHOI, a zero-shot framework to synthesize articulated human–object interactions (HOI) from monocular video priors without 3D/4D supervision. Two key ideas make this possible: (1) flow-based part segmentation that uses optical flow plus SAM masks to separate moving from static object regions and to assign Gaussians to articulated parts; and (2) a decoupled dynamics distillation pipeline that first optimizes object articulation (with kinematic/flow regularizers) and then human motion (SMPL-X) conditioned on the recovered object states using contact, smoothness, and foot-sliding losses. On benchmarks built from ArtGS and ZeroHSI-style prompts, ArtHOI improves semantic alignment (X-CLIP 0.244), contact (75.64%), penetration (0.08%), and rotation error for articulated parts (mean 6.71°) over both zero-shot and supervised baselines. Training uses 3D Gaussian splatting and reportedly runs per scene in ≈30 minutes on an A100.

**Strengths:**

- Tackles zero-shot articulated HOI—most prior zero-shot work treats objects as rigid. The decoupled optimization and the flow-->SAM-->back-projection route for part discovery are thoughtful and well-motivated.
- Clear objectives for both stages (reconstruction, tracking, articulation constraints for objects; contact/kinematics and foot-sliding for humans). The use of quasi-static pairs to stabilize articulation is neat. Ablations isolate the contribution of each loss/component.
- Consistent gains on interaction metrics (notably Contact%, Penetration%) and large reductions in articulation rotation error vs. D3D-HOI/3DADN. User study (n=51) prefers ArtHOI across criteria.
- Pipeline is easy to follow (Fig. 2/3), with explicit loss terms, implementation details, and a concise positioning table.

**Weaknesses:**

- The pipeline assumes the video diffusion prior produces correct temporal cues and rough geometry; failure modes of the prior (hallucinations, view-dependent artifacts) are not deeply analyzed. It would help to quantify robustness to flow/SAM errors.
- Many scenes are rendered or assembled from synthetic assets; it’s unclear how well the method handles in-the-wild handheld videos with clutter, motion blur, and complex backgrounds. More real-video evaluation (even without GT, with perceptual/AMT studies) would strengthen claims.
- ZeroHSI is rigid-only by design; comparing on articulated behaviors is informative but inherently favors ArtHOI. A stronger baseline would be “ZeroHSI + oracle articulation” or recent monocular articulation recovery methods plugged into ZeroHSI to test the benefit of the proposed decoupling, not just articulation presence.
- Table 5 shows identical articulation errors for w/o L_k and the full model (6.71/21.41/0.58), which seems suspicious; please double-check. Also report variance across seeds.
- ~30 minutes per scene optimization is reasonable for research, but discussion on batching, caching, or partial reuse across prompts/scenes would help for practical deployments.

**Questions:**

1. How sensitive is Stage I to optical-flow threshold τ_f and SAM prompts? Any quantitative stress test where you inject flow noise or use alternative flow estimators?
2. Can you include results on in-the-wild single-view videos with no synthetic assets? Even without GT, report X-CLIP, user studies, contact/penetration proxies.
3. Do you support multi-joint/multi-link mechanisms (e.g., cabinet with multiple doors + drawers)? Any failure cases with non-revolute joints or compound motions?
4. What happens if you feed an external monocular articulation estimator (e.g., ArticulatedGS/TAga/RIGGS-like) into a ZeroHSI-style pipeline—does the two-stage decoupling still yield an advantage?
5. Can you share optimization iteration counts vs. quality curves and opportunities for distillation to amortize per-scene optimization?
6. Please clarify the identical articulation metrics for w/o L_k vs. full model in Table 5—is this a copy error or expected?

---

### Official Review · Reviewer_BvBz · 2025-10-31

**Soundness:** 2
**Presentation:** 3
**Contribution:** 2
**Rating:** 4
**Confidence:** 2

**Summary:**

This paper presents ArtHOI, a framework that utilize the powerful foundation video models to synthesize articulated human-object interactions.
More specifically, there are two components:
1. the flow-based part segmentation which estimate the object parts (static and dynamic respectively) from the video, and
2. in the second phase, raw human motion estimation from HMR are fine-tuned conditioned on the reconstructed object states.

For this project, video foundation models (Kling) are used to generate the synthetic raw datasets.

**Strengths:**

1. The paper is well-written and provides a solid section of ablation studies.

2. The paper studies a very interesting and curical problem that lies between animation and physical AI.
Manipulation remains a task that's unsolved in both domains.
If the algorithm is proven to be scalable, we will have a rich source of datasets.

3. User studies with 51 participants shows clear perceptual advantages over the baselines.
And the ablation studies are adequate and provide insights into the contribution of each component.

**Weaknesses:**

1. The experiemnts seems to be very limited in number of scene, number of objects, and number of characters.
This raise concerns about the generalization and the robustness of the proposed method.
And there's no tool-use interactions or cases where the objects are not static in world frame (for example picking an object from a table),
which also limits the downstream applications.

2. The video and scene reconstruction quality could be improved.
It would be very helpful if the authors could provide access to the raw generated videos for a clearer comparison (along in the demos I assume).
The current scene in the attached demo appears to have significant artifacts.
Identifying and addressing the source of these scene quality issues is important, especially for downstream applications such as robotics VLA models.

3. There is a very heavy use of optimization methods accross different stages of the algorithm.
It make sthe pipeline overally complicated and very likely to be unstable, unscalable and sensitive.
Unless the authors could show that the setup can be very general accross different synthetic video input,
it will be significantly limiting its real-life applications.

To me the greatest concerns is the scalability and the robustness of the proposed method.

**Questions:**

1. why do the authors use KLing as apposed to some other similar models like Qwen / WAN
Have you experimented with other video generation models and is the proposed method sensitive to the choice of video models?
Does the algorithm works with in-the-wild videos for example from youtube, if no source 3D assets are available?

2. How well does the proposed method work with non-articulated objects?
I think it will be fair to compared against some of the method which are not designed for articulated objects such as FoundationPose.

3. What if there are two rotation joints for the articulated object? How does the algorithm handle this?
Or cases where the rotation happens in place? A typical example is rotating a lid in robotics manipulation tasks.

4. How does it handle multiple objects in the scene?

---

### Note · Authors · 2025-11-13

I have read and agree with the venue's withdrawal policy on behalf of myself and my co-authors.